# Exploring the impact of virtual reality-based mathematics learning on students' motivation: Protocol for a systematic review and meta-analysis

**Shally Novita** [1,2*], **Hari Setyowibowo**[1], **Puspita Adhi Kusuma Wijayanti**[1], **Wina Erwina**[3], **Whisnu Yudiana** [1,4], **Fredrick Dermawan Purba** [1], **Afra Hafny Noer**[1]

**1** Center for Psychological Innovation and Research, Faculty of Psychology, Universitas Padjadjaran, Sumedang, West Java, Indonesia, **2** Center for Psychometric Study, Faculty of Psychology, Universitas Padjadjaran, Sumedang, West Java, Indonesia **3** Faculty of Communication, Universitas Padjadjaran, Sumedang, West Java, Indonesia, **4** Donders Institute for Brain, Cognition and Behaviour, Radboud University Nijmegen, Nijmegen, The Netherlands

\* s.novita@unpad.ac.id

## Abstract

It has been extensively documented that motivation plays a pivotal role in both the learning and performance of mathematics, often intersecting with various antecedents of mathematical competence, such as math anxiety and self-esteem. These factors, in turn, significantly influence the well-being of students. Therefore, it is necessary to provide learning situations that may impact students' motivation in learning mathematics. During the digital era, studies have examined the use of technology, particularly, virtual reality-based mathematics learning in explaining the variance of students' motivation in mathematics learning. However, the results seem to be various and a comprehensive review about the impact of virtual reality mathematics learning on students' motivation does not seem to exist yet. This review aims to fill this knowledge gap by examining the impact of mathematics learning using virtual reality-based learning materials on students' motivation. The literature search will be conducted across multiple bibliographic databases, including Cambridge, Oxford, PubMed, Science Direct, Scopus, SpringerLink. Only studies published in English, German, or Indonesian within the past seven years will be eligible for inclusion in the review. The risk of bias inherent in this review will be evaluated by two independent reviewers utilizing the Joanna Briggs Institute (JBI) critical assessment tool. Any discrepancies between the reviewers will be resolved through the third reviewer. Further, quality of evidence will be examined using The Grading of Recommendations, Assessment, Development, and Evaluations (GRADE). A review exploring the correlation between mathematics education utilizing virtual reality and student motivation will offer educational practitioners and stakeholder's valuable insights into the effective integration of technology-based learning resources, particularly virtual reality. The findings of this study aim to elucidate the advantages and potential drawbacks associated with the adoption of virtual reality in mathematics education, as well as identify the student and instructional material characteristics that may be associated with specific increases

**Data availability statement:** No datasets were generated or analysed during the current study. All relevant data from this study will be made available upon study completion.

**Funding:** This systematic review is conducted within project of Mathematics Assessment using Virtual Reality in Indonesian Elementary School Children that was funded by Indonesian Ministry of Education with contract number 5413/UN6.3.1/PT.00/2021. However, this study protocol did not undergo any review process before submitted to Plos One.The funders had no role in study design, data collection and analysis, decision to publish, or preparation of the manuscript.

**Competing interests:** The authors have declared that no competing interests exist.

in motivation. This protocol has been registered in PROSPERO with registration number: CRD42023463974.

## Introduction

Motivation plays a crucial role in mathematics learning, as evidenced by its correlation with mathematical performance [1,2] and its interaction with key predictors such as math anxiety and self-esteem [3]. Derived from the Latin word "movere," meaning "to move" [2,4] motivation is considered a primary factor that drives and energizes human behaviour [4].

In the literature on motivation, needs are regarded as specific instances that potentially guide behaviour and possess distinct characteristics compared to goals [5]. In the context of mathematics education, students may identify a need for competence as a goal to proficiently solve tasks, a social need as a goal to contribute meaningfully to group projects, and a need for autonomy as a goal to challenge the teacher's authority [5]. The recognition of these needs as goals during mathematics learning is heavily influenced by students' beliefs, the mathematics curriculum, learning environment, and socio-mathematical norms within the classroom [5].

In the quest to identify effective strategies for enhancing motivation in mathematics learning, previous studies have investigated various interventions, including teacher interventions [6], classroom discourse interventions [7] and technology-based programs, such as the utilization of virtual reality [8]. Students in the 21st century, often characterized as digital natives, exhibit preferences for digital tools, possess advanced proficiency in information technologies [9]. Drawing from experiences with platforms like Minecraft and Roblox [10,11] traditional learning methods such as structured textbooks and linear lectures may be perceived as unengaging and unworthy of attention [12]. Consequently, digital natives have developed distinct interests, learning approaches, and lifestyles compared to their predecessors. Hence, one effective strategy to captivate the interest of digital natives and enhance their curiosity, enjoyment, and ultimately motivation is by utilizing familiar language and tools.

Virtual reality enables users to experience a near-realistic simulation, be in spaces and conditions that are impossible to be encountered in the real world, experience a high-risk environment that may be danger to face in the real world [13], as well as repeating procedural steps as many times as required according to individual learning progress, which may not be feasible in the real world [14].

Virtual reality may also help students understand abstract concepts and reduce students' misunderstanding [15]. It improves student's ability to understand abstract concepts by making those concepts more concrete, bringing them to a real-world situation, and establishing a deeper understanding [16]. For example, utilizing virtual reality for geometry instruction creates an immersive learning environment that boosts student motivation by encouraging exploration, visualization of abstract concepts, and active engagement [17].

However, while the use of virtual reality for learning holds promise in enhancing students' understanding, it also entails certain limitations and concerns. Firstly, there are health considerations for users. It is possible that during virtual session, some individuals experience physical strains such as balance disruptions and nausea(see also [18]). Secondly, there is the issue of users' technological literacy. The lack of familiarity with virtual reality technology, coupled with limited technical support, may hinder students from deriving maximum benefit from its educational applications in science [19]. Thirdly, there is the challenge of accurately replicating real-life sensory experiences and physical interactions within virtual environments [18]. While virtual reality can offer immersive and interactive experiences, it may not fully capture the complexity and authenticity of real-world encounters [18].

The objective of this systematic review is to provide a comprehensive understanding of the correlation between mathematics learning facilitated by virtual reality-based materials and student motivation in learning this subject. Specifically, this review will address following questions:

1. What is the impact of virtual reality-based mathematics learning on students' motivation?

2. What are the overlapping and discrete characteristics of virtual reality-based mathematics learning that have an impact on students' motivation?

3. Which participant characteristics may influence student motivation in the context of virtual reality-based mathematics learning?

## Methods

This review aims to examine the impact of virtual reality-based learning materials on students' motivation in learning mathematics and has been officially registered in PROSPERO under the registration number: CRD42023463974. Adhering to the guidelines set forth by the Preferred Reporting Items for Systematic Reviews and Meta-Analyses (PRISMA) that can be reviewed in the supplementary file included in this protocol [20,21,22]. During the review process, no similar review topic was identified during the review process within both PROSPERO and the Cochrane Handbook for Systematic Reviews.

### Eligibility criteria

The studies to be included in this review must meet the following criteria (Table 1):

1) participants of the studies were students aged 7-15 years old, who are still learning mathematics in elementary or secondary schools (spanning from the beginning of elementary school through the end of high school); 2) the virtual reality-based learning materials including both virtual reality headset and sticks or only headset; 3) the studies must involve mathematics learning without restriction to specific sub-domains (e.g., numerical, algebra, measurement, data analysis, geometry, etc.); 4) Studies must compare motivation levels between different participant groups (i.e., virtual reality group vs. non-virtual reality groups); 5) the studies treated motivation as an outcome and this construct was measured using instruments that were proven to have good psychometric properties (e.g., questionnaire with proven validity and reliability for measuring motivation); 6) study designs include experimental and quasi experimental studies that used quantitative and mixed-method analytical approaches; 7) studies were conducted in educational settings such as schools or child centre; 8) studies were published in English, German, or Indonesian; 9) studies were published between 2015 and 2025.

**Table 1. PICOS.**

|  | Inclusion | Exclusion |
|---|---|---|
| Population | Students aged 7-15 years old, who are still learning mathematics in elementary or secondary schools (spanning from the beginning of elementary school through the end of high school) | Individuals with mental health or specific difficulties such as: down syndrome, mentally retarded, autism, slow learner |
| Intervention | Virtual reality for mathematics learning | Other learning tools that do not include virtual reality |
| Comparison | Other learning tools such as augmented reality, computer-based learning or paper-based learning tools | – |
| Outcome | Motivation to learn mathematics | – |
| Study Design | Experimental and quasi experimental studies that used quantitative and mixed-method analytical approaches | – |

The exclusion criteria for the studies under review are as follows: 1) studies involving students with disabilities, including down syndrome, intellectual disabilities, autism spectrum disorders, slow learners, dyslexia and dyscalculia; 2) knowledge synthesis studies, such as literature reviews, scoping reviews, and systematic reviews will be excluded from consideration.

## Search strategy

The following databases will be employed to search for relevant studies: EBSCO, PubMed, ScienceDirect, Scopus, and Springer. Keywords will be tailored for each information database (refer to Table 2). All studies published between January 1st, 2015, and February 28th, 2025, will be gathered. The search strategy and procedure were devised collaboratively by the research team in conjunction with the librarian. Citation search of included studies will be conducted.

## Study selection

The data screening will be conducted by two reviewers using the open-source Rayyan application. The screening procedure included three stages: 1) removal of excluded studies, 2) title and abstract screening, and 3) full-text screening. In all stages, both reviewers will be blinded. The screening will commence with the identification and removal of duplicates, followed by an assessment of the appropriateness of publication year, study designs, language, and types of studies (e.g., reviews or book chapters), as provided by Rayyan. In the second stage, both reviewers will assess the titles and abstracts of the remaining studies and determine their inclusion based on predefined criteria. Studies meeting the inclusion criteria in this stage will proceed to the third stage for full-text screening. Any discrepancies between the reviewers will be resolved through consultation with a third reviewer.

## Data extraction

Two reviewers will extract data from the selected studies using a standardized extraction form, which will undergo pilot testing to ensure reliability [23]. This form consists of following

**Table 2. Data sources and search.**

| Database | Query |
|---|---|
| Cambridge | children OR child OR "7-15 years old" OR teenager OR adolescents OR student OR "K-12" OR "K12" AND "virtual reality" OR vr OR "virtual environment" AND learning OR education AND math OR mathematics AND motivation OR motive OR enjoyment with years filter from 2015-2025 |
| Oxford | (children OR child OR "7-15 years old" OR teenager OR adolescents OR student OR "K-12" OR "K12") AND ("virtual reality" OR vr OR "virtual environment") AND (learning OR education OR assessment OR test) AND (math OR mathematics) AND (motivation OR motive OR enjoyment) |
| Pubmed | (((((children OR child OR "7-15 years old" OR teenager OR adolescents OR student OR "K-12" OR "K12") AND ("virtual reality" OR vr OR "virtual environment")) AND (learning OR education)) AND (math OR mathematics)) AND (motivation OR motive OR enjoyment) AND years filter from 2015 - 2025 |
| Science direct | (child OR "7-15 years old" OR adolescents OR student OR "K-12" OR "K12") AND ("virtual reality" OR "virtual environment") AND (learning OR education) AND (math) with years filter 2015-2025 |
| Scopus | TITLE-ABS-KEY (children OR child OR "7-15 years old" OR teenager OR adolescents OR student OR "K-12" OR "K12") AND TITLE-ABS-KEY ("virtual reality" OR vr OR "virtual environment") AND TITLE-ABS-KEY (learning OR education) AND TITLE-ABS-KEY (math OR mathematics) AND TITLE-ABS-KEY (motivation OR motive OR enjoyment) AND PUBYEAR > 2014 AND PUBYEAR < 2026 |
| SpringerLink | (children OR child OR "7-15 years old" OR teenager OR adolescents OR student OR "K-12" OR "K12") AND ("virtual reality" OR vr OR "virtual environment") AND (learning OR education) AND (math OR mathematics) AND (motivation OR motive OR enjoyment) with years filter 2015-2025 |

aspects: 1) author name(s); 2) year of publication; 3) sample characteristics including age, gender, and, if available, information regarding socioeconomic status. Additionally, the frequency of prior digital tool usage, including virtual reality, may be recorded, as variability in familiarity with virtual reality could influence tool preferences; 4) virtual reality tools used in mathematics learning, whether only virtual reality or virtual reality with other tools and the use of complete virtual reality sets such as headset and sticks or only headset; 5) duration of virtual reality-based mathematics learning sessions; 6) the specification of mathematics materials implemented in virtual reality; 7) Conception of motivation, classified into four elements as outlined in the Background section: goal orientation, intrinsic and extrinsic motivation, interest, and self-schema [24]; 8) description of how motivation was measured; 9) study location, 10) relevant findings including any additional constructs related to motivation.

The information regarding the conception of motivation, the method of measuring motivation, and other pertinent findings elucidating the primary outcome of motivation will be utilized to investigate the first hypothesis regarding the influence of virtual reality-based mathematics learning on student motivation. Given the multifaceted nature of motivation, discerning which elements of motivation are associated with virtual reality-based mathematics learning is of significant advantage. To address the second research question, data on the specific virtual reality tools employed, the duration of mathematics learning sessions, and the materials implemented within the virtual reality environment are essential.

Finally, both sample characteristics and study location will be utilized to address the third research question, which seeks to elucidate the significance of students' characteristics in explaining motivation.

## Quality assessment

The quality of the included studies will be evaluated by two independent reviewers using the critical appraisal tool issued by the Joanna Briggs Institute (JBI) [25]. This tool comprises eight to 12 items, contingent upon study design, aimed at assessing the quality of the included studies. Each reviewer will be provided with four possible responses for each item: yes, no, unclear, and not applicable. Any discrepancies between the reviewers will be resolved through consensus and if not possible a third reviewer will be assigned.

The Grading of Recommendations, Assessment, Development, and Evaluations (GRADE) framework will be utilized to assess the quality of evidence from the included studies [26]. The GRADE framework includes assessment of risk of bias (within JBI framework it refers to critical appraisal [27]), inconsistency, indirectness, imprecision, and publication bias [28]. Two reviewers will use the GRADE to examine the quality of evidence from the pooled studies. Any disagreements between the two reviewers will be resolved through a consensus. If the consensus cannot be reached, a third reviewer will be assigned.

## Data synthesis

Both random-effects meta-analyses and the inverse variance method will be conducted. The impact of virtual reality-based mathematics learning will be quantified using a standardized mean difference (SMD) and its corresponding 95% confidence interval (95% CI). Cohen's d effect size guidelines [29] will be employed to interpret the SMD scores (0.2 = small, 0.5 = moderate, 0.8 = large difference). Graphical analysis will be conducted using forest plots to assess the heterogeneity of the studies [30].

A moderator analysis will be conducted to explore the influence of moderating variables (e.g., sample size and methodological characteristics) on the effect size and heterogeneity of the included studies. Heterogeneity will be assessed using the $I^2$ statistic, which quantifies

heterogeneity as a percentage. $I^2$ scores of < 30% indicate low heterogeneity, scores of 30–60% suggest moderate heterogeneity, and scores of > 60% indicate high heterogeneity [31]. Additionally, narrative synthesis will be employed to investigate the effects of various factors, such as study locations, socioeconomic status, and, if feasible, students' familiarity with virtual reality, on their motivation in learning mathematics.

## Discussion

Motivation stands as a pivotal determinant of student success across academic and educational trajectories. Particularly in mathematics learning, motivation has consistently demonstrated a profound impact on mathematical performance [5,32,33] and is inversely correlated with mathematics anxiety [3]. As posited in the literature, one potential avenue for enhancing motivation in mathematics learning lies in the integration of digital learning through virtual reality [8]. Through virtual reality, students are afforded the opportunity to immerse themselves in a virtual environment, which aligns with the preferences of digital natives. Moreover, virtual reality offers the means to virtually present abstract mathematical concepts, potentially augmenting students' comprehension as well.

The primary aim of this review is to investigate the potential impact of virtual reality-based mathematics learning on students' motivation in this subject, while also exploring how specific student characteristics and elements of virtual reality-based mathematics learning may contribute to variations in motivation. This information is useful particularly for educational practitioners when planning and implementing technology-based education in a classroom setting. Additionally, this review may provide valuable support to policymakers in crafting educational policies tailored to the needs of digital native students.

In addition to following PRISMA as a review guideline, this study will review studies that publish in three languages: English, German and Indonesian. Incorporating both German and Indonesian languages will broaden the scope of the study search and selection process, ultimately enhancing the richness of the review results. Meta-analysis findings, particularly the computed effect sizes, will significantly contribute to advancing understanding in this area and enhance measurement precision, thereby reinforcing the value of this review. However, it should be noted that implementing meta-analysis may pose challenges, particularly regarding heterogeneity issues inherent in cross-sectional and observational studies. Furthermore, we acknowledge the rapid growth of virtual reality technology in recent years. Consequently, the ten-year study period may not be entirely optimal. Nevertheless, we believe that to offer a comprehensive review, a broader timeframe is warranted to include a greater number of studies.

## Supporting information

**S1 Data. PRISMA-P. PRISMA-P 2015 Checklist**
(DOCX)

## Acknowledgement

We thank following individuals for their support the writing process of protocol in form of layout design and referencing (M. Rafli Iltizamullah), proofreading (Karisma Putri Diyana), and consultation of search code determination (Ziani Marni).

## Author contributions

**Conceptualization:** Shally Novita, Puspita Adhi Kusuma Wijayanti.

**Data curation:** Shally Novita, Whisnu Yudiana.

**Formal analysis:** Shally Novita, Puspita Adhi Kusuma Wijayanti, Wina Erwina, Whisnu Yudiana.

**Funding acquisition:** Shally Novita, Fredrick Dermawan Purba, Afra Hafny Noer.

**Investigation:** Shally Novita, Puspita Adhi Kusuma Wijayanti.

**Methodology:** Shally Novita, Hari Setyowibowo.

**Project administration:** Puspita Adhi Kusuma Wijayanti, Fredrick Dermawan Purba, Afra Hafny Noer.

**Supervision:** Hari Setyowibowo.

**Writing – original draft:** Shally Novita, Puspita Adhi Kusuma Wijayanti.

**Writing – review & editing:** Hari Setyowibowo.

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
