## [Decision Letter · Decision Letter 0]

29 Jan 2025

PONE-D-24-41535Exploring the Impact of Virtual Reality-Based Mathematics Learning on Student's Motivation: Protocol for A Systematic Review and Meta-analysisPLOS ONE

Dear Dr. Novita,

Thank you for submitting your manuscript to PLOS ONE. After careful consideration, we feel that it has merit but does not fully meet PLOS ONE’s publication criteria as it currently stands. Therefore, we invite you to submit a revised version of the manuscript that addresses the points raised during the review process.

**ACADEMIC EDITOR: ** Please address all of my concerns before I consider accepting this manuscript==============================

We look forward to receiving your revised manuscript.

Kind regards,

Ziyu Qi

Academic Editor

PLOS ONE

Journal Requirements:

https://www.mdpi.com/2079-9292/12/13/2832

In your revision ensure you cite all your sources (including your own works), and quote or rephrase any duplicated text outside the methods section. Further consideration is dependent on these concerns being addressed.

“Fredrick D. Purba received the Funding from the Ministry of Education. However, this study was not the main study listed in the funding proposal and therefore, the study protocol has not been externally reviewed.“

**Additional Editor Comments:**

Please address all of my concerns before I consider accepting this manuscript.

Reviewers' comments:

Reviewer's Responses to Questions

**Comments to the Author**

1. Does the manuscript provide a valid rationale for the proposed study, with clearly identified and justified research questions?

Reviewer #1: Yes

2. Is the protocol technically sound and planned in a manner that will lead to a meaningful outcome and allow testing the stated hypotheses?

Reviewer #1: Yes

3. Is the methodology feasible and described in sufficient detail to allow the work to be replicable?

Reviewer #1: Yes

4. Have the authors described where all data underlying the findings will be made available when the study is complete?

Reviewer #1: No

5. Is the manuscript presented in an intelligible fashion and written in standard English?

Reviewer #1: Yes

6. Review Comments to the Author

You may also provide optional suggestions and comments to authors that they might find helpful in planning their study.

Reviewer #1: The protocol is well organized, with clear objectives and reasonable methods. I suggest the authors consider the following aspects.

Introduction

1 Although it is commendable that this section is very detailed, it should be shortened to less than 400 words and no more than 700 words.

2 It should be stated why the study spans 2015-2024. It is recommended to review existing systematic reviews on related topics in the Introduction through a table to strengthen the rationale of the protocol. (See doi.org/10.3390/bioengineering11080741)

Methods

3 Describe the PICOS criteria in a table (still referencing the same article)

4 Search strategy should integrate the terms "K-12" or "K12" to reduce search misses

5 Citation search of included articles should be performed (forward or backward snowballing method).

6 Please update the archive on the PROSPERO webpage to always keep it consistent with this article

7 Authors should describe where all data underlying the findings will be made available when the study is complete.

Minor corrections

8 PRISMA should be cited in the text. (10.7326/0003-4819-151-4-200908180-00135)

7. PLOS authors have the option to publish the peer review history of their article (what does this mean? ). If published, this will include your full peer review and any attached files.

**Do you want your identity to be public for this peer review?** For information about this choice, including consent withdrawal, please see our Privacy Policy .

Reviewer #1: **Yes: ** Ziyu Qi

---

## [Author Response · Author response to Decision Letter 1]

4 Feb 2025

Response to the reviewer

Reviewer #1: The protocol is well organized, with clear objectives and reasonable methods. I suggest the authors consider the following aspects.

Author(s): Thank you.

Introduction

1 Although it is commendable that this section is very detailed, it should be shortened to less than 400 words and no more than 700 words.

Author(s): the actual introduction consists of 629 words (between 400 and 700 words).

2 It should be stated why the study spans 2015-2024. It is recommended to review existing systematic reviews on related topics in the Introduction through a table to strengthen the rationale of the protocol. (See doi.org/10.3390/bioengineering11080741)

Author(s): this is a mistake due to the extended deadline based on internal circumstances. We have extended the time range of study publication into 2025.

Methods

3 Describe the PICOS criteria in a table (still referencing the same article)

Author(s): a PICOS table is included in the manuscript

4 Search strategy should integrate the terms "K-12" or "K12" to reduce search misses

Author(s): both terms have been included in the revised search strategy.

5 Citation search of included articles should be performed (forward or backward snowballing method).

Author(s): we included this information in the search strategy section

6 Please update the archive on the PROSPERO webpage to always keep it consistent with this article

Author(s): we will update the title registration in the PROSPERO after peer review process is completed.

7 Authors should describe where all data underlying the findings will be made available when the study is complete.

Author(s): this information has been submitted as metadata. Since this protocol does not include any pilot data, we stated “data are not reported” in this manuscript. However, we included a statement regarding the plan to publish the data once the systematic review is completed: “Since this manuscript is a study protocol, the data will be published in Open Science Framework upon publication of the systematic review and meta-analysis paper” (please check our responses to the following instruction from the Plos One’s system: Describe where the data may be found in full sentences. If you are copying our sample text, replace any instances of XXX with the appropriate details.)

Minor corrections

8 PRISMA should be cited in the text. (10.7326/0003-4819-151-4-200908180-00135)

Author(s): thank you for this detail feedback. We cited PRISMA as reference list [20] to [22].

7. PLOS authors have the option to publish the peer review history of their article (what does this mean?). If published, this will include your full peer review and any attached files.

Author(s): Yes, we would like to publish the peer review history.

---

## [Decision Letter · Decision Letter 1]

6 Feb 2025

Exploring the Impact of Virtual Reality-Based Mathematics Learning on Student's Motivation: Protocol for A Systematic Review and Meta-analysis

PONE-D-24-41535R1

Dear Dr. Novita,

We’re pleased to inform you that your manuscript has been judged scientifically suitable for publication and will be formally accepted for publication once it meets all outstanding technical requirements.

Kind regards,

Ziyu Qi

Academic Editor

PLOS ONE

Additional Editor Comments (optional):

The manuscript can be accepted in its current form.

Reviewers' comments:

Reviewer's Responses to Questions

**Comments to the Author**

1. Does the manuscript provide a valid rationale for the proposed study, with clearly identified and justified research questions?

Reviewer #1: Yes

2. Is the protocol technically sound and planned in a manner that will lead to a meaningful outcome and allow testing the stated hypotheses?

Reviewer #1: Yes

3. Is the methodology feasible and described in sufficient detail to allow the work to be replicable?

Reviewer #1: Yes

4. Have the authors described where all data underlying the findings will be made available when the study is complete?

Reviewer #1: Yes

5. Is the manuscript presented in an intelligible fashion and written in standard English?

Reviewer #1: Yes

6. Review Comments to the Author

You may also provide optional suggestions and comments to authors that they might find helpful in planning their study.

Reviewer #1: I am satisfied with the author's revisions and rebuttals and have no further concerns or comments. This manuscript can be accepted in its presenting form.

7. PLOS authors have the option to publish the peer review history of their article (what does this mean? ). If published, this will include your full peer review and any attached files.

**Do you want your identity to be public for this peer review?** For information about this choice, including consent withdrawal, please see our Privacy Policy .

Reviewer #1: **Yes: ** Ziyu Qi

---

## [Editor Report · Acceptance letter]

PONE-D-24-41535R1

PLOS ONE

Dear Dr. Novita,

I'm pleased to inform you that your manuscript has been deemed suitable for publication in PLOS ONE. Congratulations! Your manuscript is now being handed over to our production team.

Kind regards,

on behalf of

Mr. Ziyu Qi

Academic Editor

PLOS ONE